# TRAINING ACTIVE VISION REASONERS WITH MULTI-TURN REINFORCEMENT LEARNING

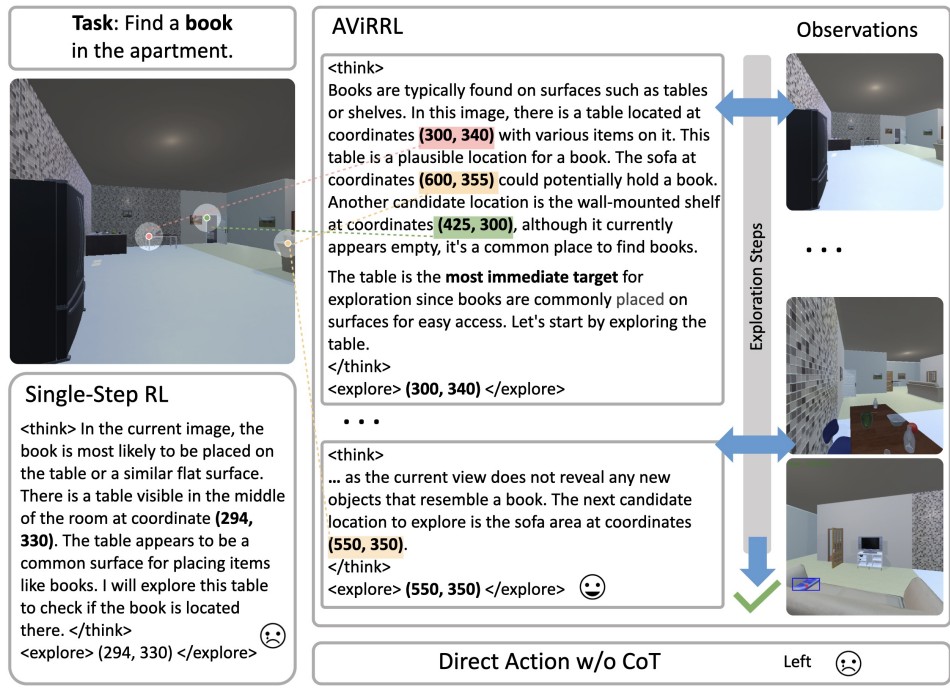

Figure 1: **Active visual reasoning enables interpretable, long-horizon exploration.** AViRRL performs long-term reasoning to efficiently search 3D environments and localize objects. Its reasoning stays temporally consistent: after not finding books on the table, it revisits the sofa seen previously. In contrast, single-step behavior cloning reasons myopically.

## ABSTRACT

Vision-Language Models (VLMs) face a critical challenge in active vision: determining where to look in 3D environments to answer questions or locate objects. While reinforcement learning (RL) over chains of thought has boosted performance in visual tasks with static inputs like passive image understanding, extending this reasoning-based approach to active vision remains an open challenge. We introduce AViRRL—Active Visual Reasoning with Reinforcement Learning—a multi-turn reinforcement learning approach to train vision-language models to acquire reasoning strategies for active visual exploration. Our approach combines two key components: (1) a tree-search-guided data generation method for warm starting active vision reasoning strategies in VLMs, and (2) online multi-turn reinforcement learning that optimizes full thought–action trajectories based on task success. On TinyNav, a benchmark for visual search of small objects in realistic 3D environments, we show that our method significantly outperforms both visual navigation prompting methods and single-step reinforcement learning with behavior cloning rewards. Compared to baselines, our agent exhibits more effective active vision behaviors, including exploration efficiency and reasoning aligned with visual state. Our work shows that reinforcement learning on thought-action trajectories enables VLMs to develop active visual reasoning, shifting from passive perception to autonomous, goal-directed 3D exploration.

## 1 INTRODUCTION

Active vision, the ability to intelligently control perception through purposeful camera movements and exploration in 3D visual environments, represents a fundamental challenge in embodied AI. While humans coordinate eye movements, head turns, and navigation to gather visual information, artificial agents struggle with this seemingly basic capability. The predominant paradigm for active vision has been to train reactive models that directly map observations to actions, bypassing explicit reasoning about what to look for and where (Zhang et al., 2024b;a; Zeng et al., 2024a). However, this kind of "reactive" control is not necessarily the best choice in settings that demand broad generalization, such as novel scenes. In such situations, a person might need to think more carefully. If they were asked to locate specific items in a cluttered room, they might first try to figure out likely locations based on context and object relationships, rather than simply performing a learned search pattern from "muscle memory." In the same way, we would like our active vision policies to both perform well-practiced exploration strategies, and to "reason through" novel situations before grounding their commands into actions. Such reasoning might include identifying and locating task-relevant objects, producing a plan to accomplish visual search, and considering multiple possible actions before choosing one.

Recent advances in vision-language models (VLMs) have demonstrated that chain-of-thought reasoning can decompose complex visual understanding into interpretable steps (Campbell et al., 2024), and that reinforcement learning over these reasoning chains or tool calls improves performance on passive visual tasks (Sarch et al., 2025b; Wang et al., 2025b). However, extending this reasoning-based approach to active vision—where models must reason not just about what they see but about where to look next—remains an open challenge. Most approaches for training reasoning for VLM agent tasks focus on 2D visual environments like GUI navigation (Wang et al., 2025a) or train for a single time step of reasoning and action using behavior cloning rewards (Liu et al., 2025a; Wu et al., 2025). This leads us to pose the research question: **How can we train VLM reasoners that perform multi-step active vision in 3D environments, moving beyond passive observation and single-step reasoning?**

To address these challenges, we extend the RL paradigm from passive perception tasks to optimization of reasoning-action trajectories in active visual reasoning—coupling reasoning VLMs with interactive environments (Kolve et al., 2017; Savva et al., 2019). We propose AViRRL—Active Visual Reasoning with Reinforcement Learning—a two-stage framework that explicitly learns visual reasoning behaviors for active vision in 3D environments.

First, existing VLMs exhibit limited zero-shot proficiency in reasoning about where and how to move its camera to locate an object, thus requiring a warm start to establish a foundational understanding of reasoning behaviors for active vision. We address this through a Monte Carlo Tree Search (MCTS) warm-start approach that systematically explores reasoning-action outputs. By using detailed prompts that guide scene-grounded state estimation and exploration, we generate high-quality thought-action trajectories suitable for multi-turn supervised fine-tuning. This warm-start strategy constrains the search space during online RL.

Second, we apply online multi-turn reinforcement learning using Group Relative Policy Optimization (GRPO) (Shao et al.) to optimize full thought-action trajectories, enabling the model to learn coordinated reasoning and camera actions over entire exploration episodes. Crucially, this approach generates supervision without any human annotations or curated reasoning data, allowing visual reasoning strategies to emerge through sparse reward and interaction with the environment. To support large-scale training, we developed an asynchronous, parallel simulation system for GRPO with VLMs in simulators, executing 128 concurrent rollouts on a server with eight A6000 GPUs.

We evaluate our approach on TinyNav, a new benchmark for embodied visual search that requires models to locate small objects in realistic 3D environments. Our method demonstrates strong performance across two distinct settings. In the in-view setting, where objects are visible but extremely small, AViRRL-3b achieves 20% higher pass@1 accuracy than ViGoRL (Sarch et al., 2025a)-3b, a state-of-the-art passive reasoning VLM, demonstrating that our active viewpoint reasoning outperforms passive VLM reasoning. Additionally, our 3B parameter model outperforms SOTA zero-shot navigation baselines (Goetting et al., 2024; Zhao et al., 2024) by 11% pass@1.

In the more challenging out-of-view setting, where objects are initially outside the camera's field of view, AViRRL-3b surpasses reinforcement learning methods that use behavior cloning rewards for learning reasoning and action, an approach adopted by recent VLM navigation works (Liu et al.,

2025a; Wu et al., 2025), by 14% pass@1. Through ablation studies, we demonstrate that both our MCTS warm start and online RL components are crucial for achieving these performance gains. Our behavioral analysis reveals that our model exhibits more effective active vision behaviors, including object exploration efficiency and reasoning aligned with visual state, which contribute to the higher task success rates. Our results indicate that reinforcement learning over thought-action trajectories allows VLMs to develop active visual reasoning capabilities, shifting them from passive perception to autonomous, goal-directed exploration of 3D environments.

## 2 RELATED WORK

**Active Vision Policies.** Standard approaches to visual navigation policy learning typically involve training neural networks from scratch using collected trajectory datasets (Shah et al., 2021b;a; Chang et al., 2023; Zheng et al., 2022; Mao et al., 2023). This paradigm is limited by datascale and generalizability. Recent works increasingly pursue the development of universal navigation models capable of generalizing across diverse scenarios. PoliFormer (Zeng et al., 2024b) operate on RGB observations with natural language objectives, offering greater practical applicability in real-world deployments. In parallel, Move to See Better (Fang et al., 2021) introduces a self-improving embodied object detection framework, repositioning agents to enhance perspective. SEAL (Chaplot et al., 2021) further develops self-supervised embodied active learning, leveraging exploration and 3D consistency. Nevertheless, these models train reactive policies from scratch to map observations directly to actions. In contrast, we learn on top of pretrained VLMs to map observations to reasoning and actions.

**VLMs for Active Vision.** Recent efforts (Brohan et al., 2022; Zitkovich et al., 2023; Kim et al., 2024) have explored adapting pre-trained vision-language models (Bai et al., 2025; Driess et al., 2023) through domain-specific fine-tuning on visual navigation demonstrations. While this strategy can yield more robust policies compared to training from scratch, the supervised fine-tuning process can limit generalization capabilities. Another branch of work investigates leveraging zero-shot pre-trained VLMs by designing appropriate prompting strategies that encourage task-aligned outputs. Methods like VLFM (Yokoyama et al., 2024; Gadre et al., 2022) employ visual encoders like CLIP to bridge visual observations with linguistic goal descriptions. Recent works prompts VLMs to give step-by-step navigation actions based on visual inputs (Yu et al., 2023; Kuang et al., 2024; Yin et al., 2024; Bhorkar, 2024). VLMnav (Goetting et al., 2024) converts the navigation problem to a sequence of question-answering tasks by annotating images with possible moving directions. In contrast, our approach fine-tunes pre-trained VLMs with reinforcement learning, directly optimizing over thought-action trajectories.

**Reasoning in VLMs.** Prompting methods encourage VLMs to reason through explicit intermediate steps (Wei et al., 2022; Kojima et al., 2023; Zhang et al., 2024c; Zhou et al., 2024; Wu et al., 2024), often by decomposing tasks into sub-components (Wu & Xie, 2023; Hu et al., 2024a; Gupta & Kembhavi, 2022; Fu et al., 2025; Surís et al., 2023), though they rely on fixed architectures and handcrafted prompts. Supervised training embeds reasoning via distillation (Xu et al., 2025), search-generated trajectories (Sarch et al., 2024; Hu et al., 2024b; Yao et al., 2024), or grounding fine-tuning (Li et al., 2025). Reinforcement learning, effective where outcomes are verifiable (Jaech et al., 2024; Guo et al., 2025; Team et al., 2025), is being explored for visual reasoning: Visual-RFT (Liu et al., 2025b) applies RL only to text, while ViGoRL (Sarch et al., 2025a) integrates visual interaction for exploration. Nav-R1 (Liu et al., 2025a) advances this with a dual-system "Fast-in-Slow" paradigm, trained on large-scale CoT and refined with GRPO-based single-turn RL rewards. Yet most methods still optimize single-step reasoning, risking shortcuts, whereas we optimize thought-action episodes.

## 3 PRELIMINARIES

**Task Setup: Embodied Active Visual Search** We formalize category-conditioned active visual search in 3D simulators as a finite-horizon POMDP $\mathcal{M} = \langle \mathcal{S}, \mathcal{A}, \mathcal{O}, T, \Omega, r, \gamma, \rho_0 \rangle$ with horizon $T_{\max}$. At initialization, a scene $s_1 \sim \rho_0$ is sampled, a target category $c \in \mathcal{C}$ is provided, and the agent spawns at a valid pose. The goal is to localize at least one instance of category $c$ and output a valid 2D point in the egocentric view that lies within that instance.

At each timestep $t$, the agent receives observation $o_t = (I_t^{\mathrm{rgb}}, c) \in \mathcal{O}$, where $I_t^{\mathrm{rgb}} \in \mathbb{R}^{H \times W \times 3}$ is the RGB image. We employ a hybrid action space $\mathcal{A} = \mathcal{A}_{\mathrm{nav}}^{\mathrm{low}} \cup \mathcal{A}_{\mathrm{nav}}^{\mathrm{point}} \cup \mathcal{A}_{\mathrm{det}}$ comprising: (i) low-level

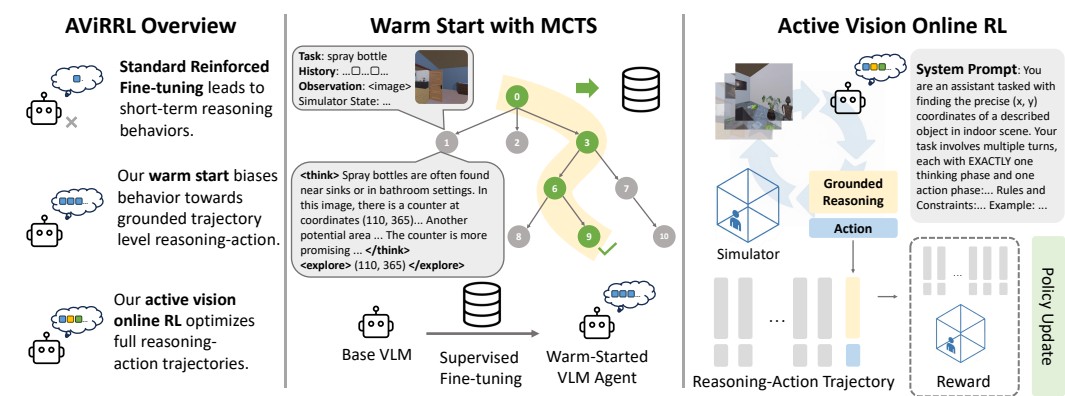

Figure 2: The overview of our method. **Left:** Instead of standard reinforcement finetuning, we develop a two-stage training pipeline with MCTS Warm Start and Active Vision Online RL. **Mid:** In MCTS Warm Start, we extract high-quality reasoning-action trajectories from MCTS on a large VLM, for supervised fine-tuning of a smaller student VLM. **Right:** In Active Vision Online RL, we further optimize the student VLM with GRPO on the trajectory level, where the agent learns goal-directed long-term reasoning and action.

navigation FORWARD($\delta$), TURNLEFT($\theta$), TURNRIGHT($\theta$) with fixed step size $\delta$ and rotation $\theta$; (ii) point-to-move navigation GOTO($u, v$) for $(u, v) \in [0, 1]^2$ that projects normalized image coordinates to 3D waypoints and executes motion using the raycasting function provided by simulator; and (iii) detection DETECT($p$) that predicts a point $p = (x, y) \in [0, 1]^2$ and terminates the episode.

Success occurs when the predicted point falls within any ground-truth bounding box $b^\star \in \mathcal{B}t(c)$ for the target category. We use sparse terminal reward: $r_t = R_{\text{succ}}$ for success, otherwise $r_t = 0$.

**Problem Setup: Learning to Reason and Act** We learn an active visual reasoning policy that interleaves latent textual reasoning with environment actions. At each timestep, the policy emits both a latent "thought" string $z_t \in \Sigma^*$ (delimited by <think>...</think> tags) and an action $a_t \in \mathcal{A}$:
$$\pi_\theta(a_t, z_t \mid h_t, c) \quad \text{with} \quad h_t = (o_{1:t}, a_{1:t-1}).$$

Only $a_t$ affects the environment; $z_t$ serves as a latent variable that structures the action distribution. The objective maximizes expected discounted return: $J(\theta) = \mathbb{E}s_1 \sim \rho_0, \tau \sim \pi\theta[\sum_{t=1}^T \gamma^{t-1} r_t]$.

# 4 AViRRL: Active Visual Reasoning with Reinforcement Learning

This section introduces our proposed AViRRL framework, a multi-turn reinforcement learning approach to train VLMs to acquire reasoning strategies for active visual exploration. An overview of our method is shown in Figure 2. First, we warm-start via supervised fine-tuning on MCTS-generated trajectories that include structured reasoning about scene state and exploration proposals (Section 4.1). Second, we apply trajectory-level reinforcement learning using Group Relative Policy Optimization (GRPO) (Section 4.2).

## 4.1 Tree-Search-Guided Warm Start

We extract reasoning and action priors from a large VLM (Qwen2.5-VL-72B) using Monte Carlo Tree Search (MCTS) (Figure 2; Middle), where the VLM serves as the actor and the simulator provides environmental feedback. The search tree maintains nodes representing environment states (egocentric observations with corresponding simulator states) and edges representing actions from our hybrid action space $\mathcal{A}$.

During expansion, we select nodes with highest UCB scores. At each selected node, the VLM processes the observation history $h_t$ and current image $I_t^{\text{rgb}}$, generating multiple candidate pairs of thoughts $z_t$ and actions $a_t$. Each pair is executed in the simulator to create child nodes. Terminal

rewards propagate back through the tree to update node statistics. A tree succeeds if any leaf node achieves successful detection of the target category $c$.

**Visually-Grounded Scene Analysis.** When processing observations, we prompt the VLM to generate structured reasoning within '<think>...</think>' tags before selecting actions. This reasoning includes: (i) identification of visible objects with pixel coordinates following coordinate-based grounding formats, and (ii) spatial relationship analysis between observed objects and the target category. This explicit reasoning leverages the VLM's pre-trained commonsense knowledge about object co-occurrences and typical spatial layouts, informing subsequent navigation decisions through either point-to-move $\text{GOTO}(u, v)$ or low-level navigation actions.

**Trajectory Extraction.** From successful MCTS trees, we extract the highest-reward path from root to successful detection. This path is linearized into a sequence of $(o_t, z_t, a_t)$ tuples, creating demonstration trajectories $\mathcal{D}_{\text{ws}}$ where thoughts and actions are VLM-generated and simulator-verified. This automated process requires no human annotation, enabling scalable data collection.

**Supervised Fine-Tuning.** We train a smaller student VLM (Qwen2.5-VL-3B) on these trajectories using the SFT objective from our problem formulation. The model learns to predict both thought tokens and action tokens conditioned on observation history, minimizing cross-entropy loss: $\mathcal{L}_{\text{SFT}}(\theta) = -\sum_t [\log \pi_\theta(z_t \mid h_t) + \log \pi_\theta(a_t \mid h_t, z_t)]$. This warm-start phase transfers the large VLM's reasoning patterns to our policy $\pi_\theta$, establishing strong priors for spatial reasoning and exploration strategies before trajectory-level RL refinement.

## 4.2 ONLINE REINFORCEMENT LEARNING FOR ACTIVE VISION REASONING

After warm-starting with MCTS-generated trajectories, we refine the policy through multi-turn reinforcement learning that directly optimizes for task success (Figure 2; Right). This phase addresses the distribution shift between teacher-generated demonstrations and the student's own trajectory distribution, while strengthening the connection between reasoning content and navigation outcomes.

**Multi-turn Navigation.** We frame each episode as a sequence of navigation decisions followed by a terminal detection action. At each timestep, the policy generates both reasoning $z_t$ and an action from our hybrid space—either low-level navigation, point-to-move $\text{GOTO}(u,v)$, or terminal $\text{DETECT}(p)$. The multi-turn structure allows the agent to iteratively refine its beliefs about object locations through exploration, with each observation potentially revealing new visual cues that inform subsequent reasoning. During training, we sample trajectory groups from the current policy and compute group-relative advantages (Shao et al.), enabling the model to learn from contrasts between successful and unsuccessful search strategies within the same scene.

**Trajectory-Level Reasoning Reward Design.** Our reinforcement learning uses three complementary reward signals evaluated at the trajectory level. The **success reward** reflects task success—whether the final detection point falls within the target object's bounding box. The **format reward** ensures proper output structure, penalizing malformed commands or invalid action parameters. The **distance bonus**, used by previous work (Zeng et al., 2024a), encourages approaching the target by rewarding reductions in distance to the nearest target instance (e.g., moving from 1.5m to 1.0m yields a proportional reward), while not penalizing exploration that temporarily increases distance. This asymmetric distance reward promotes efficient convergence toward targets without discouraging necessary exploratory movements. We provide more details of our reward design in Appendix **??**.

**Asynchronous Parallelized Simulation** We develop an asynchronous, parallel simulator orchestration to efficiently scale RL with VLMs. First, to avoid disproportionate overhead from per-episode scene initialization and agent resets, we bind each rollout to a dedicated, persistent simulator process that serves the same scene across steps, eliminating repeated re-initialization. Second, to mitigate stragglers arising from heterogeneous episode lengths during multi-turn training, we perform task-aware grouping: rollouts from the same task—empirically exhibiting similar horizons—are assigned to the same simulator pool to maximize parallel utilization. Finally, we incorporate fault-tolerant mechanisms that maintain simulator liveness and automatically restart crashed processes, ensuring robustness under large-scale parallelization. These designs enable scaling to 128 concurrent rollouts on a single machine with 8 A6000 GPUs. More implementation details will be released with our codebase.

We provide additional implementation details in Appendix A2, A3, and A4.

## 5 EXPERIMENTS

We evaluate our model, AViRRL, on category-conditioned visual search in 3D environments. We investigate the following research questions:

- **RQ1:** How does AViRRL perform on active vision tasks?
- **RQ2:** How important is each component of AViRRL?
- **RQ3:** Why is AViRRL effective at active vision?

### 5.1 EVALUATION SETUP

**Benchmark.** We introduce TinyNav, a benchmark for embodied category-conditioned visual search in realistic 3D indoor environments. TinyNav is built on ProcTHOR (Deitke et al., 2022), which procedurally generates diverse apartment layouts and object placements. We focus on small, frequently occluded objects (e.g., cup, pencil, remote control) that require fine-grained viewpoint control. TinyNav comprises two regimes: (i) *Object-in-View*, where the target is initially visible but small, emphasizing precise approach and localization; and (ii) *Object-out-of-View*, where the target is initially not visible, emphasizing exploration and long-horizon reasoning. Because ProcTHOR supports unbounded procedural generation, we can synthesize arbitrarily large training sets; the exact number of training scenes per model is reported in the Appendix. For evaluation, we use a held-out test split of 100 Object-in-View and 200 Object-out-of-View episodes.

**Episode protocol.** Each episode provides a high-level instruction (e.g., "find the cup in the apartment"). At each step, the agent observes the current RGB frame, the full interaction history, and the instruction, and selects one action from the hybrid action space (Section 3). The simulator executes the action and returns the next observation; invalid actions yield an error signal. Episodes terminate when the agent issues DETECT($p$) or the step budget is exhausted. For Object Navigation, success is counted if DETECT is issued while the target is in view and within 2.0 meters, following Poliformer (Zeng et al., 2024a). For Object Detection, success requires that $p$ lies within a ground-truth bounding box of the target category in the final frame. We report Pass@k, counting an episode as successful if the agent solves the task within k attempts.

### 5.2 RQ1: IMPORTANCE OF REASONING IN ACTIVE VISION

**Baselines.** To test the effectiveness of our method, we compare with the following categories:

(1) *Large proprietary or open-sourced VLMs with prompting*: general purpose large VLMs including Gemini-2.5-Pro (Team et al., 2024) accessed with API and Qwen2.5-VL-72B-instruct (Bai et al., 2025) deployed locally. We prompt these models with 1) grounded reasoning prompt, which introduces the action space and encourages the model to reason on the observation and history, with an example. (Attached in the Appendix.) and 2) visual prompting methods, Scene Imagination from ImagineNav (Zhao et al., 2024), and Action Proposer from VLMnav (Goetting et al., 2024). We provide full details of baseline setups in Appendix A5.

(2) *Single-turn Behavior Cloning*: These methods trains a Qwen-3B, the same base model we use for our method, with behavior cloning under the grounded reasoning prompt. MCTS-SFT, which trains the model to clone reasoning and actions from trajectories collected from MCTS. Behavior Cloning with GRPO (GRPO-BC) (Liu et al., 2025a), which trains the model to mimic actions from MCTS trajectories. ViGoRL (Sarch et al., 2025b), which trains the model to reason about the current observation and output the answer directly without moving around in the scene. We provide full details of baseline setups in Appendix A5 and A2.

(3) *Zero-shot GRPO*: Under the grounded reasoning prompt, we follow (Guo et al., 2025) to apply GRPO directly on a Qwen-3B VLM without injecting reasoning priors.

Results of these experiments are shown in Table 1 and Table 2. Our model consistently outperforms all baselines by a large margin on both Object-in-View and Object-out-of-View settings. We summarize the key findings from these experiments as follows.

**AViRRL beats strong prompted VLMs and behavior cloning on TinyNav.** On Object-in-View, our unified model achieves 48.0% Pass@1 (Navigation) and 59.0% Pass@1 (Detection), outperforming

Table 1: Performance on object-out-of-view setting.[†]: evaluated using their main contribution without mapping components. We highlight the best and second-best results in bold and underline respectively. Our method outperforms all baselines by a large margin in all metric.

| Method | Model | Object Navigation | | | Object Detection | | |
|---|---|---|---|---|---|---|---|
| | | Pass@1 | Pass@3 | Pass@10 | Pass@1 | Pass@3 | Pass@10 |
| *Zero-shot Prompting* | | | | | | | |
| Grounded Reasoning | Qwen2.5-VL-72B | 13.0 | 18.5 | 30.5 | 10.0 | 17.5 | 35.0 |
| Grounded Reasoning | Gemini-2.5-Pro | 12.5 | 17.5 | 23.5 | 3.5 | 11.0 | 18.0 |
| Grounded Reasoning | Qwen2.5-VL-3B | 6.0 | 14.0 | 22.5 | 0.5 | 1.0 | 3.0 |
| ImagineNav (Zhao et al., 2024)[†] | Gemini-2.5-Pro | 7.0 | 9.5 | 12.5 | 2.5 | 7.5 | 14.0 |
| VLMnav (Goetting et al., 2024)[†] | Qwen2.5-VL-72B | 22.5 | 33.0 | 49.0 | 17.5 | 23.0 | 44.0 |
| *Single-turn RL* | | | | | | | |
| GRPO-BC-No-Thinking | Qwen2.5-VL-3B | 8.0 | 12.5 | 22.5 | 3.0 | 10.0 | 23.0 |
| GRPO-BC-Thinking | Qwen2.5-VL-3B | 13.0 | 15.0 | 17.0 | 27.0 | 31.0 | 38.0 |
| *Ours* | | | | | | | |
| **AViRRL (Ours)** | Qwen2.5-VL-3B | **27.0** | **39.0** | **58.5** | **34.5** | **47.5** | **63.5** |
| *Ablations* | | | | | | | |
| Warm Start Only (w/o RL) | Qwen2.5-VL-3B | 12.5 | 19.0 | 28.5 | 9.0 | 19.5 | 26.5 |
| RL w/o Warm-Start | Qwen2.5-VL-3B | 6.0 | 15.5 | 27.5 | 4.5 | 14.5 | 27.5 |

Table 2: Performance on in-view setting. *: ViGoRL is trained with our dataset. ViGoRL does not do physical action so we do not report its Object Navigation results. [†]: evaluated using their prompting method without mapping components. We highlight the best and second-best results in bold and underline respectively. Our method outperforms all baselines by a large margin in Pass@1.

| Method | Model | Object Navigation | | | Object Detection | | |
|---|---|---|---|---|---|---|---|
| | | Pass@1 | Pass@3 | Pass@10 | Pass@1 | Pass@3 | Pass@10 |
| *Zero-shot Prompting* | | | | | | | |
| Grounded Reasoning | Qwen2.5-VL-72B | 41.0 | 60.0 | 71.0 | 31.0 | 56.0 | 74.0 |
| Grounded Reasoning | Gemini-2.5-Pro | 30.0 | 51.0 | 64.0 | 13.0 | 32.0 | 50.0 |
| Grounded Reasoning | Qwen2.5-VL-3B | 7.0 | 17.0 | 28.0 | 4.0 | 8.0 | 20.0 |
| ImagineNav (Zhao et al., 2024)[†] | Gemini-2.5-Pro | 8.0 | 11.0 | 17.0 | 5.0 | 19.0 | 41.0 |
| VLMnav (Goetting et al., 2024)[†] | Qwen2.5-VL-72B | 37.0 | 54.0 | **78.0** | 39.0 | 56.0 | 72.0 |
| *Single-turn RL* | | | | | | | |
| ViGoRL(Sarch et al., 2025b)* | Qwen2.5-VL-3B | - | - | - | 46.0 | 60.0 | 77.0 |
| *Ours* | | | | | | | |
| **AViRRL (Ours)** | Qwen2.5-VL-3B | **48.0** | **63.0** | 73.0 | **59.0** | 67.0 | 79.0 |
| *Ablations* | | | | | | | |
| Warm Start Only (w/o RL) | Qwen2.5-VL-3B | 22.0 | 38.0 | 58.0 | 44.0 | **69.0** | **86.0** |
| RL w/o Warm-Start | Qwen2.5-VL-3B | 9.0 | 22.0 | 48.0 | 8.0 | 15.0 | 42.0 |

the best baselines, Qwen2.5-VL-72B and ViGoRL, by 7.0% and 13.0% respectively. On Object-out-of-View, our model achieves 27.0% Pass@1 (Navigation) and 34.5% Pass@1 (Detection), outperforming the best baselines, VLMnav and GRPO-BC-Thinking, by 4.5% and 7.5%.

**Trained small models outperform large prompted models.** Existing zero-shot VLMs or prompting methods exhibit limited active vision abilities. Compared with Gemini-2.5-Pro and Qwen2.5-VL-72B under recent prompting approaches for visual navigation (Zhao et al., 2024; Goetting et al., 2024), our trained small model outperforms large models in both settings: +7.0% and +4.5% in Object Navigation (in-view and out-of-view), and +20.0% and +17.0% in Object Detection (in-view and out-of-view).

**Our trajectory-level reasoning is more effective than learning step-wise reasoning with behavior cloning rewards.** Reasoning plays a crucial role in active vision, as shown by the contrast between GRPO-BC-Thinking and GRPO-BC-No-Thinking. In particular, reasoning over entire trajectories proves more effective than step-wise reasoning. Compared with single-step behavior cloning (GRPO-BC-Thinking) (Liu et al., 2025a), our method cultivates trajectory-level reasoning and achieves a 10.8% average improvement in pass@1. This suggests that warm-starting multi-step active vision

Table 3: Pattern analysis on Object-out-of-view setting. SEL means the Success weighted by Episode Length. Higher is better for all metrics. Our AViRRL achieves competitive performance across all metrics.

| Method | Exploration Effectiveness | Grounding Alignment | SEL |
|---|---|---|---|
| Qwen2.5-VL-72B | 3.2% | 0.981 | 11.7% |
| Gemini-2.5-pro | 4.4% | 0.984 | 6.0% |
| Qwen-3B | 2.3% | 0.235 | 0.4% |
| Warm Start Only | 2.4% | 0.31 | 5.9% |
| RL w/o Warm-Start | 2.1% | 0.880 | 0.5% |
| GRPO-BC-Thinking | 3.1% | 0.1 | 19.3% |
| AViRRL (Ours) | 3.7% | 0.964 | 22.1% |

behaviors and reinforcement learning over full trajectories—rather than isolated steps—leads to stronger performance. Section 5.4 and the visualizations in the Appendix further highlight the coherence and effectiveness of trajectory-level reasoning.

**Active viewpoint reasoning beats passive reasoning.** Against ViGoRL (Sarch et al., 2025a), a SOTA approach for learning static reasoning without moving, our approach that explicitly moves and reasons about where to look yields a +13.0% improvement.

**The injection of reasoning priors significantly boosts the performance of small models.** From the comparison between Zero-shot and MCTS-SFT, we found that introducing reasoning priors through tree search based supervised fine-tuning significantly boosts the performance of small models, achieving a difference between 4% and 44% in Object Detection in Object-in-View setting, and a difference between 0.5% and 9.0% in Object Navigation in Object-out-of-View setting. This demonstrates the effectiveness of reasoning priors in enhancing the capabilities of smaller models.

## 5.3 RQ2: ABLATION STUDY OF AViRRL

We ablate AViRRL's two-stage training to quantify the contributions of (i) warm-starting with reasoning priors (MCTS-SFT) and (ii) trajectory-level reinforcement learning with GRPO.

**Warm-starting with reasoning priors is essential.** Removing the MCTS-SFT warm start ("RL w/o Warm-Start"), i.e., applying GRPO directly to a Qwen-3B VLM, reduces accuracy by an average of 35.3% compared to our full model, yielding only a minor 2.5% gain over the base model. This indicates that without reasoning priors, RL alone fails to learn effective policies.

**Trajectory-level GRPO amplifies reasoning priors.** Removing GRPO ("Warm Start Only"), i.e., training with MCTS-SFT but no RL, decreases accuracy by an average of 20.3% relative to our full model. With a strong prior from MCTS-SFT, GRPO further refines step-level supervision into consistent trajectory-level policies, yielding robust improvements across tasks. Additional trajectory-level analyses are provided in Section 5.4.

## 5.4 RQ3: EXPLORATION PATTERN ANALYSIS

We analyze several exploration and reasoning patterns with corresponding metrics to interpret why AViRRL is effective. The patterns and metrics are summarized in Table 3.

**Pattern 1: Exploration Effectiveness.** Exploration effectiveness refers to the agent's ability to efficiently explore the environment and perceive previously unseen areas. The Exploration Effectiveness $S_{EE}$ metric is defined as $S_{EE} = N_{observes}/(N_{total} \times (t-1))$, where $N_{observes}$ is the number of instances observed by the agent (excluding the initial view), $N_{total}$ is the total number of instances in the environment, and $t$ is the total number of steps taken. A higher $S_{EE}$ indicates more effective exploration, i.e., the agent observes more unique instances per step.

From the table, our model demonstrates competitive $S_{EE}$ scores, comparable with Gemini-2.5-Pro and showing substantial improvement over the 3B ablation models. Specifically, our model shows a 1.3% absolute (54% relative) improvement over Warm Start Only, and a 0.6% absolute (19% relative)

advantage over GRPO-BC-Thinking. These results indicate that training trajectory-level reasoning with GRPO improves exploration efficiency and the agent's ability to gather crucial information for active vision tasks.

**Pattern 2: Grounding Alignment.** Grounding Alignment measures how well the model's grounded thinking corresponds to the actual target in the environment. The Grounding Alignment $S_{GA}$ metric is computed as the average feedback score from large language models (LLMs) assessing the model's grounded reasoning at each step against the actual target object. Feedback scores range from 0.0 to 1.0, with higher values indicating better alignment. Results show that our model attains a near-best $S_{GA}$, only 0.02 lower than Gemini-2.5-Pro but with significantly higher accuracy, and substantially outperforms the 3B ablation models. This suggests the model learns goal-directed reasoning that better aligns with the visual state compared to ablated versions of our model. We provide examples of high and low scoring grounding alignment in Appendix A5.

**Pattern 3: Success weighted by Episode Length.** We report Success weighted by Episode Length (SEL) to measure the efficiency of task completion, following (Zeng et al., 2024a). $S_{\text{SEL}} = (1/N) \sum_{i=1}^{N} S_i (w_i / \max(\varepsilon, e_i))$, where $N$ is the total number of episodes, $S_i \in \{0, 1\}$ indicates success for episode $i$, $e_i$ is the episode length for episode $i$, $w_i$ denotes the shortest number of steps to find the target (e.g., set to 1 for successful episodes and 0 for unsuccessful ones), and $\varepsilon$ is a small constant to avoid division by zero. This metric rewards agents that succeed in fewer steps, reflecting more efficient task completion. We compute $w_i$ by performing a breadth-first search from the ground-truth target position on the simulator map. From the table, our model achieves the highest SEL score, outperforming all baselines and ablations. This indicates more effective and efficient exploration, resulting in faster task completion.

## 6 DISCUSSION - CONCLUSION

We present AViRRL, a two-stage training approach that combines a grounded MCTS warm start with online multi-turn reinforcement learning to optimize full thought–action trajectories for active visual reasoning in 3D environments. Evaluated on TinyNav, AViRRL substantially outperforms prompted large VLMs and single-step RL baselines, while ablations show both the MCTS warm start and trajectory-level RL are essential. Behavioral analyses reveal improved exploration efficiency, action consistency, and reasoning alignment, demonstrating that reinforcement learning over thought–action trajectories transforms VLMs from passive observers into autonomous, goal-directed explorers.

**Limitations and Future Work.** While our MCTS warm-start requires substantial computational resources during data collection, this investment yields an efficient 3B model. Future work could explore more efficient reasoning prior generation through smaller teacher models or synthetic data. Our evaluation focuses on category-conditioned indoor small object search. Extending the evaluation to more complex queries, dynamic environments, or manipulation tasks would further test generalization. Finally, while our behavioral analyses demonstrate that AViRRL develops coherent exploration strategies, the precise mechanisms linking reasoning to action should be further investigated through interventions to add or remove certain reasoning and acting behaviors.

**Ethics Statement.** This research uses only synthetic environments with procedurally generated scenes from ProcTHOR, involving no human subjects. We acknowledge potential biases toward North American home layouts and English-only inputs. Our TinyNav benchmark contains only common household objects without sensitive content. While our method improves visual search in simulation, real-world deployment would require additional safety and privacy considerations beyond this work's scope. We will release all code and models under permissive licenses. We have no conflicts of interest to declare.

**Reproducibility Statement.** We provide comprehensive implementation details to ensure reproducibility: hybrid action space and reward specifications (Section 3, Appendix A), MCTS procedures and prompts (Section 4.1, Appendix B), GRPO training configuration (Section 4.2, Appendix C), and TinyNav benchmark details built on open-source ProcTHOR. We will release our complete codebase including model checkpoints, evaluation scripts, MCTS trajectory generation pipeline, and distributed training infrastructure upon acceptance. All experiments use publicly available Qwen2.5-VL base models. Training requires 8 A6000 GPUs with specific compute details in Appendix D.

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

# A APPENDIX

## A1 APPENDIX OVERVIEW

The structure of this Appendix is as follows:

- Section A2 contains training details including dataset distribution, model training procedures, and hyperparameter configurations.
- Section A3 contains optimization and reward design details including the GRPO objective and reward composition.
- Section A4 contains MCTS warm start configuration including teacher model settings and trajectory extraction procedures.
- Section A5 contains evaluation settings, prompts, and qualitative results with grounding alignment examples.

## A2 TRAINING DETAILS

### A2.1 TRAINING SET AND DATA DISTRIBUTION

We collect 35k training episodes for both Object-in-view and Object-out-of-view settings. Each episode contains 2–18 steps. The dataset is split into 31k episodes for training and 4k for validation.

To ensure fairness of comparison, we strictly control the total number of training samples across all models. The settings for each variant are:

- **AViRRL (Ours)**: 16k trajectories for SFT and 15k prompts for GRPO
- **GRPO-BC-Thinking**: 31k trajectories for single-step GRPO
- **GRPO-BC-No-Thinking**: 31k trajectories for single-step GRPO
- **Warm start only**: 31k trajectories for SFT
- **RL w/o Warm-Start**: 31k prompts for GRPO

### A2.2 MODEL TRAINING PROCEDURE

#### A2.2.1 AViRRL (OURS)

We first perform supervised fine-tuning (SFT) on the collected MCTS trajectories. Training took ~4 hours on 8 L40S GPUs.

We then apply GRPO to further polish reasoning ability. GRPO training took ~1.5 days on 8 L40S GPUs.

**GRPO-only** and **SFT-only** variants follow the same procedure, with only the respective stage applied.

#### A2.2.2 GRPO-BC VARIANTS

For GRPO-BC-Thinking, we follow the same training procedure as AViRRL, except we use each step's action in an MCTS trajectory as the target action, instead of verifying trajectory correctness via the simulator. The reward at each step is composed as:

$$r = 0.5 \times \text{action match} + 0.5 \times \text{format reward}.$$

The format reward is 1 if the output includes `<think>` and `<explore>` or `<answer>` tags.

For GRPO-BC-No-Thinking, we drop the requirement for `<think>` tags and remove thinking prompts from the system message. The reward reduces to schema checks without validating reasoning tags:

$$\text{format reward} = 1 \text{ if output contains } \texttt{<explore>} \text{ or } \texttt{<answer>}.$$

### A2.3 TRAINING HYPERPARAMETERS

Student models are Qwen2.5-VL-3B. Vision tower frozen unless noted. Mixed precision bf16.

Table A1: Supervised fine-tuning (SFT) hyperparameters.

| Hyperparameter | Value |
|---|---|
| Epochs | 3 |
| Learning rate | $1 \times 10^{-6}$ |
| Weight decay | 0.01 |
| Warmup ratio | 0.03 |
| Batch size | 8 |
| Grad. accumulation | 4 |
| Effective batch size | 32 |
| Scheduler | Cosine |
| Precision | bf16 |
| Flash attention | fa2 |
| Freeze vision tower | True |
| Max seq. length | 8192 |
| Deepspeed config | ZeRO Stage 3 |

Table A2: GRPO hyperparameters.

| Hyperparameter | Value |
|---|---|
| Max prompt length | 7168 |
| Max response length | 4096 |
| Max generation per turn | 1024 |
| Max turns | 5 |
| Observation size | 616×616 |
| Learning rate | $1 \times 10^{-6}$ |
| KL coefficient | 0.01 |
| Weight decay | 0.01 |
| Warmup ratio | 0.05 |
| Clip ratio | 0.28 |
| Grad. clipping | Max norm 0.2 |
| Rollout batch / grad batch | 64 / 32 |
| Training steps | 40 |
| Group size | 16 |
| Rollout engine | vLLM |
| Top-$p$ / Temp. | 0.99 / 1.0 |
| Mixed precision | bf16 |
| Freeze vision tower | True |

## A3 OPTIMIZATION AND REWARD DESIGN

### A3.1 GRPO OBJECTIVE AND TOKEN-LEVEL CREDIT ASSIGNMENT

We use Group Relative Policy Optimization (GRPO) with group-normalized advantages and clipped token updates. For $G$ trajectories $\{\tau^{(i)}\}_{i=1}^{G}$ with returns $r^{(i)}$ and group mean $\bar{R} = \frac{1}{G}\sum_i r^{(i)}$, define $\hat{A}^{(i)} = r^{(i)} - \bar{R}$. Over output tokens (thought *and* action) $\tau_t^{(i)}$,

$$\mathcal{L}_{\text{GRPO}}(\theta) = -\frac{1}{G}\sum_{i=1}^{G}\frac{1}{|\tau^{(i)}|}\sum_t \min\left[\rho_t^{(i)}\,\hat{A}^{(i)},\ \text{clip}\big(\rho_t^{(i)},\,1-\varepsilon,\,1+\varepsilon\big)\,\hat{A}^{(i)}\right] + \beta\,\text{KL}[\pi_\theta \,\|\, \pi_{\text{ref}}],$$

with $\rho_t^{(i)} = \dfrac{\pi_\theta(\tau_t^{(i)} \mid \tau_{<t}^{(i)}, x)}{\pi_{\text{old}}(\tau_t^{(i)} \mid \tau_{<t}^{(i)}, x)}$. Observation tokens are excluded; malformed/non-terminated samples are masked.

### A3.2 REWARD COMPOSITION

Trajectory-level reward is a convex combination of success, format, and progress:

$$R(\tau) = \alpha_{\text{succ}}\,\mathbf{1}\{\text{success}\} + \alpha_{\text{fmt}}\,\text{Format}(\tau) + \alpha_{\text{prog}}\,\text{Progress}(\tau).$$

The overall trajectory reward is composed as:

$$R = 0.875 \times \text{success} + 0.125 \times \text{format} + 0.1 \times \text{progress},$$

where:

- **Success:** final DETECT point inside a relaxed ground-truth box (binary). The relaxed bounding box is the ground truth 2D bounding box simulator provided with boundaries extended 30 pixels in each direction (i.e. $[x_{min} - 30, x_{max} + 30] \times [y_{min} - 30, y_{max} + 30]$).
- **Format:** schema checks for well-formed outputs (ordered `<think>...</think>` and one valid action per step).
- **Progress:** asymmetric shaping rewarding approach without penalizing exploratory detours:

$$\text{Progress}(\tau) = \sum_{t=1}^{T-1} \max\big(0, \ d_t - d_{t+1}\big),$$

where $d_t$ is the simulator distance to the nearest target at step $t$ (equals $0.1\times$ meters moved toward target).

We keep $\alpha$ values fixed across experiments (not tuned per dataset).

## A4  MCTS WARM START CONFIGURATION

**Teacher and search.** We use Qwen2.5-VL-72B-Instruct as the actor during search. Nodes store value/visits; selection uses UCB; expansion generates up to three (thought, action) candidates; rollouts simulate to terminal/depth cap; backprop updates incremental means.

**Grounded thoughts.** The teacher emits `<think>` text that (i) grounds visible objects with pixel coordinates, (ii) reasons about spatial relations/likely target locations, (iii) proposes an action valid in the hybrid action space.

**Trajectory extraction.** From each successful tree, we take the best root-to-leaf path to form $(o_t, z_t, a_t)$ tuples for SFT.

Table A3: MCTS configuration parameters.

| Hyperparameter | Value |
|---|---|
| Teacher model | Qwen2.5-VL-72B-Instruct |
| Simulations per input | 24 |
| Max tree depth | 18 |
| Rollouts per node | 2 |
| Children per expansion | 3 |
| $c_{\text{puct}}$ | 2.0 |
| Sampling temperature | 1.0 |
| Top-$p$ | 1.0 |
| Max tokens per node | 512 |
| Parallel processes | 10 |

## A5  EVALUATION AND PROMPTS

### A5.1  EVALUATION SETTINGS

For inference, the temperature is set to 0.5 across all models (Gemini, Qwen72B, Qwen3B). Input images are resized to $616 \times 616$.

### A5.2  GROUNDED REASONING PROMPT

We provide the full prompt used in our experiments in Listing 1:

Listing 1: System prompt for ViRRL

```
You are an assistant tasked with finding the precise (x, y) coordinates of a described object
    in a photo of an indoor scene.
Your task involves multiple turns, each with EXACTLY one thinking phase and one action phase:
Thinking Phase <think>...</think>:
Systematically analyze the current image to identify at least two distinct candidate regions
    where the object might be located
For each candidate region:
 Describe what is visible at that specific location
 Explain why it might match the target object
 Provide precise coordinates for that region (x, y)
Compare these candidates step-by-step, evaluating the evidence for each
Determine which candidate region is most promising for further exploration or final answering
If unclear about object identity, infer based on likely context, purpose, or typical placement
Always verify each candidate by examining multiple possibilities before making decisions
Format all coordinates as (x, y)
You should not deliberately repeat the coordinates in the history, because that is no good for
    your thinking.
Action Phase:
Based on your reasoning, output exactly ONE of the following:
1) Explore a specific region by coordinates: <explore> (x, y) </explore> - Use this to move to
    the most promising position to search for or get closer to the object.
2) Perform a low-level move: <explore>left</explore>, <explore>right</explore>, or <explore>
    forward</explore> - Use these to take a single-step movement in the specified direction
    when a directional move is more appropriate than moving to explicit coordinates.
3) Provide final answer: <answer> (x, y) </answer> - Use this only when confident you have
    located the target object.
Rules and Constraints:
Maximum of 3  <explore> actions (pointing/left/right/forward) before you MUST provide a final
    answer
 Always verify your reasoning carefully before giving a final answer
Only provide an answer if confident; otherwise, continue exploring
Ground all reasoning in explicit coordinate references when giving coordinate explores
Each candidate evaluation should describe the region first, then assess its relevance to both
    the target object and previous exploration steps
Example: (Task: Find the book.)
<think>Books are typically found on desks, bookshelves, or tables. In the current view, I can
    identify some promising locations: there appears to be a desk surface at coordinate (213,
     460) with some clustered objects that could include books, and I can see what looks like
     a bookshelf at coordinate (507, 321) with multiple rectangular objects arranged
    vertically that match typical book placement patterns. I also see a bed at coordinate
    (105, 459), but beds are typically irrelevant with books. A door can be seen at
    coordinate (380, 347), which could lead to other rooms for further exploration. Given
    that bookshelves are the most common location for books, the bookshelf region at (507,
    321) represents the higher-probability candidate for exploration. </think><explore> (507,
     321) </explore>
```

The variant for GRPO-BC-No-Thinking is provided in Listing 2:

Listing 2: System prompt for no thinking baselines

```
You are an assistant tasked with identifying precise (x,y) coordinates of a described object
    in an photo on an indoor scene.
Your task involves multiple turns of reasoning, each with EXACTLY one <think> step and one
    action:
- At each turn, first clearly reason about what you see in the photo and where the target
    object is likely to be in <think> </think> tags. Then output an action.
- The action can be one of the following:
a) Output an explore action to go to a new position, to look for or go closer to the object,
    formatted precisely as:
  <explore> (x, y) </explore>
b) If you are sure the current image does not contain the object, you should output <explore>
    return </explore> to go back to the initial view.
c) If confident you are near the object, output your final answer enclosed in <answer> (x, y)
    </answer> tags.
- You have a maximum of 15 explore actions. After 15 explorations, you MUST give a final
    answer.
Always carefully verify each exploration before providing your final answer.
- Only answer if you are confident about the answer. If you are not confident, output an
    exploration action. You should not always end after one turn.
```

## A5.3  QUALITATIVE RESULTS

**Example trajectories from AViRRL** In Figure A1 and Figure A2, we provide example trajectories from our model.

**Grounding Alignment Examples.** In Section 5.4 we compare our model and baselines on grounding alignment. Example trajectories are included:

Listing 3: Example of reasoning with high grounding alignment

```
<think>Toilet paper is typically found near the toilet, either on a holder attached to the
    wall or placed on the floor nearby. In this image, the toilet is located at coordinates
    (190, 365). There doesn't appear to be any visible toilet paper in the immediate vicinity
     of the toilet. However, since toilet paper is often stored out of sight, it's possible
    it could be located further along the wall or on the floor to the left of the toilet. The
     wall space to the left of the toilet extends from approximately (150, 300) to (200, 400)
    , which may contain a toilet paper holder or storage area. The other side of the image
    shows a wall that extends from (300, 300) to (350, 450), which may also conceal a toilet
    paper holder or storage area. The most promising location to explore first would be the
    wall space to the left of the toilet.</think><explore> (150, 300) </explore>
<think>At the current location, the wall is predominantly brown with no visible toilet paper.
    The wall extends from around (50, 50) to (550, 550). Since the toilet paper wasn't found
    here, it might be located further down the wall or on the other side of the toilet. The
    toilet is located at coordinates (100, 350), and the wall continues from there. The next
    logical step would be to explore the area to the right of the toilet.</think><explore>
    right </explore>
<think>After turning right, the wall now shows a vertical object at coordinates (400, 450)
    that resembles a toilet paper holder. This object is positioned near the edge of the
    frame, suggesting it might extend further out of view. The toilet is partially visible at
     coordinates (350, 550), with the base and part of the tank showing. The toilet paper
    holder is a plausible candidate for the toilet paper since it is a common place to find
    toilet paper. It's important to check this location closely as it aligns well with the
    typical placement of toilet paper.</think><explore> (400, 450) </explore>
<think>The object at (400, 450) is a vertical structure that appears to be a toilet paper
    holder, but there is no visible toilet paper attached to it. Given that the toilet paper
    is usually placed directly on the holder, it may be located just a short distance away
    from the holder. Since the toilet paper is not visible at this location, it might be
    located slightly lower or in another location nearby. The toilet is visible at
    coordinates (300, 500), and the wall continues from there. Exploring the lower part of
    the wall near the toilet might reveal the toilet paper.</think><explore> (300, 550) </
    explore>
<think>At the current location, the wall continues unbroken from (50, 50) to (550, 550). There
     is no sign of toilet paper in this section. Given the layout, the toilet paper might be
    located on the opposite wall or even on the floor. The toilet is partially visible at
    coordinates (300, 500), and the wall continues from there. The next logical step is to
    explore the wall to the right of the toilet.</think><explore> right </explore>
<think>The current view shows a section of the wall and a small part of the floor with a
    corner visible at coordinates (550, 500). The toilet is partially visible at coordinates
    (100, 500), and the wall continues from there. The toilet paper holder is still visible
    at coordinates (150, 450). Since the toilet paper has not been found yet, it might be
    located on the opposite wall or on the floor. Exploring the lower part of the wall near
    the toilet is a good next step.</think><explore> (100, 550) </explore>
```

Listing 4: Example of reasoning with low grounding alignment

```
<think>Given the context of finding a pan in the apartment, we need to consider common
    locations such as the kitchen or bathroom. The current image shows a bathroom with a
    toilet and a picture frame on the wall. There's no immediate indication of a pan in the
    image. Let's explore the bathroom further by moving forward.</think>
<think>This is the second image. Let's re-evaluate the possible locations for the pan. The
    current image does not show a pan, but there is a picture frame on the wall. It could be
    that the pan is in another room or not immediately visible here. Given the lack of any
    immediate evidence of a pan, I will continue exploring the bathroom further.</think>
<think>Since the pan has not been observed yet and the current image is still part of the
    bathroom, it is possible that the pan is in another part of the apartment. The shadow
    suggests movement, indicating that I may have moved to a different area. Let's move left
    to check if there is any indication of the pan.</think>
<think>The shadow indicates movement, suggesting that I have moved to a different area. The
    image shows a wall with a picture frame similar to the previous image, but now it appears
    to be a different part of the bathroom. However, there is no indication of the pan, so I
    cannot confirm its presence here. Given the lack of a pan in this area and the shadow
    indicating movement, it's possible the pan is in another part of the apartment.</think>
<think>The image shows a wall with a picture frame, which is consistent with previous
    observations. However, there is no indication of the pan in this area. The shadow again
    suggests movement, indicating that I have explored further within the same room. Since
    the pan has not been found, it is possible that it is in another part of the apartment.</
    think>
<think>Since the pan has not been found in the bathroom and the shadow suggests movement, it
    is possible that the pan is in another part of the apartment. The previous explorations
    have not revealed the pan, so it is reasonable to continue searching in the same general
    area.</think>
<think>Exploring the bathroom further did not reveal the pan. The shadow again suggests
    movement, indicating that I have moved to a different part of the bathroom. Given the
    lack of a pan and the shadow movements, it is possible that the pan is in another part of
    the apartment.</think>
<think>The pan has not been found yet, and the shadow suggests movement. It is possible that
    the pan is in another part of the apartment. The previous explorations have not revealed
    the pan, so it is reasonable to continue searching in the same general area.</think>
```

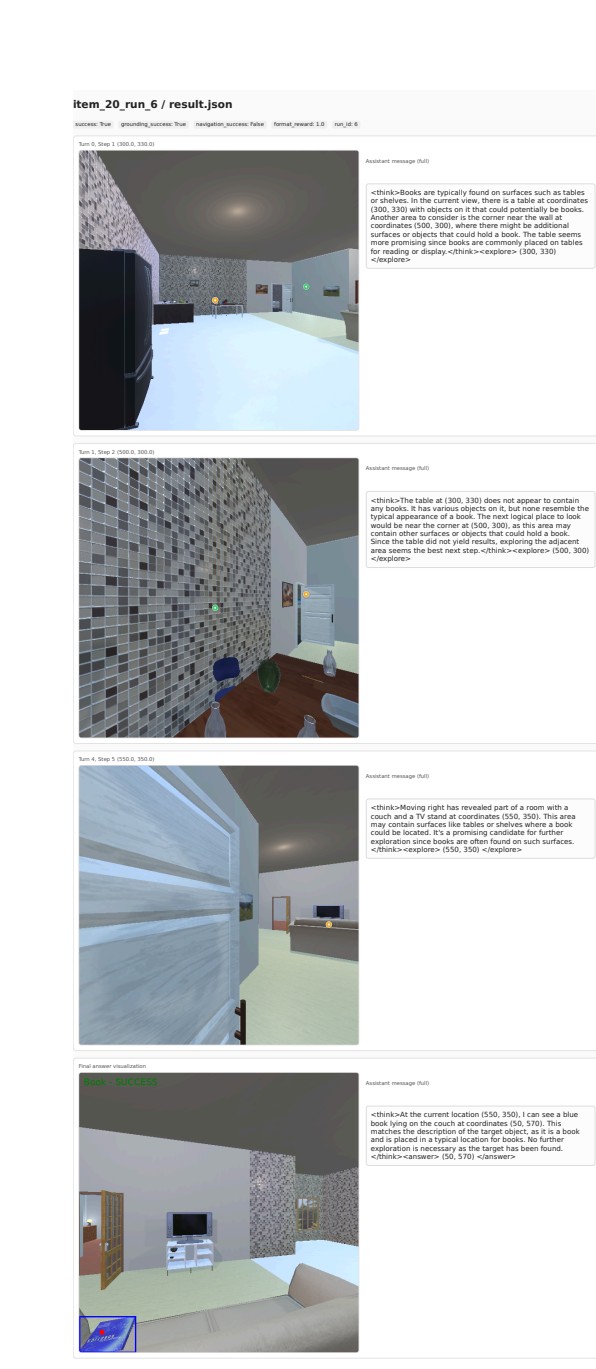

Figure A1: Example trajectory of AViRRL.

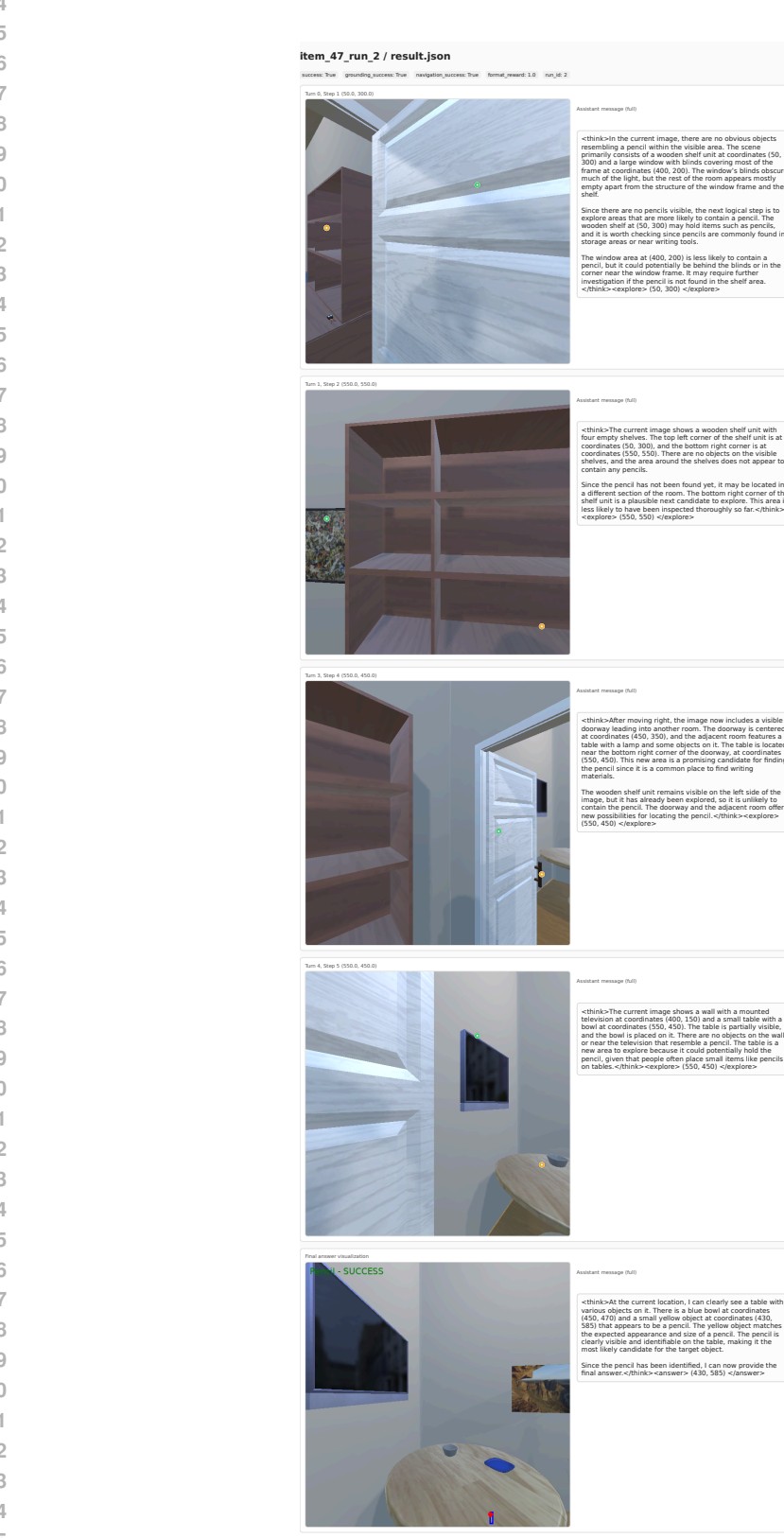

Figure A2: Example trajectory of AViRRL.

