# OpenReview forum: "Training Active Vision Reasoners with Multi-Turn Reinforcement Learning"
_ICLR.cc/2026/Conference — Submitted to ICLR 2026_

### Official Review · Reviewer_KnRF · 2025-10-25

**Soundness:** 2
**Presentation:** 2
**Contribution:** 2
**Rating:** 4
**Confidence:** 4

**Summary:**

The paper claims to contribute AVIRRL, a two-stage framework for training Vision-Language Models (VLMs) in active visual search tasks within 3D environments. The core problem identified is that existing methods use myopic, single-step reasoning, whereas this work focuses on multi-turn reasoning. The proposed method first distils high-level reasoning–action trajectories from a large teacher VLM via Monte-Carlo Tree Search (MCTS) and then refines a 3 B-parameter student with group-relative policy optimisation (GRPO) on full exploration episodes. The authors introduce a new benchmark, TinyNav , on which they claim their 3B model outperforms both large zero-shot VLMs and single-step RL baselines.

**Strengths:**

1.	Addresses a meaningful gap between passive reasoning and active, goal-directed exploration in embodied AI.
2.	Introduction of TinyNav may benefit the community as a specialized benchmark for active vision.

**Weaknesses:**

1.	Novelty is incremental relative to closely related, very recent work. ViGoRL already proposes MCTS warm-start of grounded reasoning followed by GRPO, including multi-turn RL with visual feedback. Moving from 2D spatial data to embodied 3D scenes is incremental domain transfer, not a fundamentally new algorithm.
2.	Evaluation is limited to TinyNav, a small-object search benchmark. No validation on other widely-used embodied AI benchmarks, raising questions about generalization. Besides, no comparisons to other RL-based embodied navigation baselines on mainstream datasets.
3.	Lack ablations on the reward design, such as the contribution of each reward component and the sensitivity to their respective weights.
4.	The reward formulation is essentially identical to conventional navigation rewards (success rate, path fidelity, endpoint proximity). It does not introduce any metrics or shaping signals that explicitly encourage active scene-specific exploration or reasoning behaviors — for example, rewards for discovering novel viewpoints, reducing uncertainty, or verifying scene attributes. This undermines the claim of novelty in "active visual reasoning."
5.	The warm-start trajectories come entirely from Qwen2.5-VL-72B with MCTS. This risks simply distilling the teacher’s biases and errors into the student — if the teacher performs poorly on certain exploration strategies, the student may inherit these mistakes. More ablations on the teacher model should be provided.

Typos:
1.	“Appendix ?? ” appears in Line 258.
2.	Missing explanations for some symbols in Line 156.
3.	Compute claims are inconsistent (A6000 vs L40S; 128 rollouts vs the SFT/RL timings). Please reconcile and report exact configurations for each phase.

**Questions:**

See Weaknesses

---

### Official Review · Reviewer_jCdQ · 2025-10-30

**Soundness:** 2
**Presentation:** 2
**Contribution:** 1
**Rating:** 2
**Confidence:** 4

**Summary:**

This paper studies active visual reasoning in a 3D simulator. The authors propose a two-stage pipeline for training a vision-language model (VLM) capable of performing navigation and question answering:
(i) Warm-start by sampling thought–action trajectories from a large vision-language model and using them to supervise and fine-tune a 3B student model;
(ii) Apply trajectory-level GRPO with customized rewards (based on success, response format, and distance heuristics) to optimize multi-turn active reasoning and action outputs. The proposed two-stage framework (AViRRL) is evaluated on TinyNav, a new AI2-ProcTHOR environment that supports simplified, controlled episodes for navigation-based object search tasks. Experimental results show that the proposed two-stage pipeline outperforms simple baselines.

**Strengths:**

- This paper is well-written and easy to read.
- On the proposed TinyNav benchmark, the AViRRL-trained model outperforms both prompted large vision-language models and non-interactive, single-turn RL baselines. Ablation experiments show that both the MCTS warm-start and trajectory-level GRPO contribute significantly to performance.

**Weaknesses:**

- Limited technical contribution. This paper merely repackages several well-known techniques (warm-starting, GRPO, and LLM-based reasoning) and applies them to another domain. The main novelty appears to be the introduction of the TinyNav environment, but this setting is overly simplified for studying general active or navigation-based visual reasoning. For instance, recent work has begun to explore VLM navigation in more complex environments such as Minecraft-based exploration tasks. As a result, TinyNav feels too bespoke and narrow to support claims of broad generalization.
- Weak evaluations. All evaluations are conducted solely within the proposed TinyNav environment. It remains unclear how the model would perform on standard benchmarks such as HM3D, MP3D ObjectNav, or Habitat-based tasks. Moreover, TinyNav’s action space includes high-level GOTO and DETECT APIs, whereas VLMNav models typically output low-level movement commands—making this comparison arguably unfair. The use of a distance-based reward during TinyNav training (unavailable to baseline models) further raises concerns about comparability.
- Lack of generalization discussion. The use of a VLM for navigation should, in principle, support cross-simulator, cross-dataset, or even potential sim-to-real transfer. However, the paper does not analyze how the GRPO fine-tuning process affects these generalization capabilities.

**Questions:**

How does strong RL baselines perform? e.g., PoliFormer? For baselines, what does it mean by prompting without mapping components?

---

### Official Review · Reviewer_Jcv1 · 2025-11-01

**Soundness:** 2
**Presentation:** 2
**Contribution:** 2
**Rating:** 4
**Confidence:** 5

**Summary:**

This paper introduces AViRRL, a two-stage framework that trains vision-language models (VLMs) for active visual reasoning in 3D embodied environments. The method combines (1) a Monte Carlo Tree Search (MCTS)-guided warm start, generating high-quality reasoning-action trajectories for initial supervised fine-tuning, and (2) trajectory-level online reinforcement learning using Group Relative Policy Optimization (GRPO) to optimize coordinated reasoning and camera actions over full episodes. Evaluated on the TinyNav benchmark, AViRRL demonstrates significant improvement over state-of-the-art prompting and baseline RL methods, especially for locating small objects both in- and out-of-view, and shows more effective, interpretable exploration behavior.

**Strengths:**

- **Addressing a Real Gap**: The paper tackles the under-explored challenge of extending chain-of-thought and reasoning-based RL from passive perception to active 3D visual exploration, presenting a compelling direction for embodied vision-language models.
- **Methodological Rigor & Two-Stage Framework**: The MCTS-guided warm start, followed by trajectory-level RL with GRPO, is well-motivated and grounded with practical implementation details (Section 4). Figure 2 clearly outlines the pipeline, illustrating the progressive refinement from large model outputs through supervised and RL stages.
- **Careful Experimental Evaluation**: Results presented in Table 1 (object-out-of-view) and Table 2 (in-view) show consistent improvements, often by substantial margins, across different baselines and ablation studies. For instance, AViRRL outperforms single-step RL baselines by 14% pass@1 on out-of-view detection, and small student models surpass prompted large VLMs.
- **Clear Mathematical Formulations & Implementation Transparency**: The formulation of the hybrid action space, loss functions ($\mathcal{L}_{\mathrm{SFT}},\ \mathcal{L}_{\mathrm{GRPO}}$), and reward composition are well-structured, with key implementation and code-releasing promises outlined in both the main text and Appendix.
- **System-Level Contributions**: The asynchronous and scalable parallel simulation system for RL rollouts demonstrates an impressive engineering effort, further enabling reproducibility and future scaling.

**Weaknesses:**

1. **Insufficient Comparison with Some Directly Related Multi-Turn RL and Reasoning Methods**: Despite a thorough related work section, the paper omits discussion or empirical comparisons with several very closely related efforts in multi-modal, multi-turn RL, such as works on verifier-free RL pretraining (see 'RLP: Reinforcement as a Pretraining Objective'), in-parameter latent CoT reasoning, and scalable verifier-free reasoning (see 'VFScale'). Their absence weakens the claim of comprehensive advancement over the latest in multi-turn, trajectory-level RL for reasoning—especially since some may employ similar or complementary reward, feedback, or hierarchical schemes.
2. **Limited Scope of Realistic Benchmarking**: TinyNav, although a sensible benchmark, is focused only on synthetic ProcTHOR environments and small-object search tasks. This restricts the evidence for generalization to more diverse or real-world scenarios (such as those involving dynamic environments, manipulation, or more complex spatial-temporal reasoning).
3. **Reward Shaping and Sensitivity**: Section A3 discusses the reward shaping (success, format, and progress) but lacks ablation on reward weighting or the sensitivity of the final behavior and performance to these hyperparameters. Over-reliance on distance bonuses or format rewards could potentially bias policy learning towards “gaming” these rewards rather than genuine task completion or exploration, especially in sparse-reward settings.
4. **Disconnection between Reasoning and Action Unpacking**: While reasoning outputs $z_t$ clearly structures the action space (and thought-action pairs are generated per step), there is limited quantitative analysis of the actual causal effect of different types/forms of reasoning generated by the agent on task performance. Beyond the grounding-alignment metric (Table 3), it's not clear if the RL actually incentivizes “better” reasoning patterns, or simply learns to game the reward by producing plausible-looking but shallow chains.
5. **Scalability and Efficiency of Warm Start**: The MCTS-warm start, as acknowledged in Section 6, is resource-intensive and could be prohibitive for training at scale or with more complex teacher models/environments. The potential for this pipeline to handle larger-scale settings or more temporally extended tasks is unclear.
6. **Limited Discussion of Negative and Failure Cases**: There are sparse discussions and limited concrete examples of scenarios where AViRRL fails or underperforms, especially in environments with occlusions, distractors, or non-standard object layouts.
7. **Potential Opaqueness of Trajectory-Level Reward Attribution**: Equation for $\mathcal{L}_{\mathrm{GRPO}}$ and the discussion of token-level credit assignment (Appendix A3.1) are mathematically sound, but the handling of partial credit, malformed outputs, and non-terminated samples is described in qualitative terms only. A more concrete treatment (e.g., examples or empirical consequences) would clarify robustness.
8. **Minor Writing/Presentation Issues**: The paper, while generally well-written and dense, contains small typographic errors (e.g., interpolation of latex math into tables in Table 1/2 headers; dangling references for 'Appendix ??'; and some heavy referencing). These do not fundamentally hinder comprehension but do slightly affect clarity.

### Figure and Table

- **Figure 1**: Clearly communicates the contrast between AViRRL and single-step RL in terms of long-horizon, interpretable, and temporally consistent visual exploration. However, more quantitative support for the types/frequency of such “revisitation” or “backtracking” behaviors (beyond the single visual example) would further strengthen the case.
- **Figure 2**: The schematic breaks down the methodology pipeline lucidly. However, the policy update process on the rightmost side could further detail how reward signals feed into the GRPO updates; currently, the link between “Grounded Reasoning Action” and “Policy Update” is somewhat visually abstract.
- **Tables 1, 2**: Showcase competitive performance across multiple baselines and ablations. A potential concern is the lack of statistical significance measures or confidence intervals, and the absence of certain strong baselines (e.g., from the latest multi-turn RL for multimodal reasoning works). Also, while the separation between navigation and detection metrics is clear, joint metrics or compounded task results could provide deeper insights.
- **Table 3**: The use of Exploration Effectiveness, Grounding Alignment, and SEL is insightful. The calculation of SEL, which weights success by episode length, meaningfully demonstrates improved efficiency. However, the methodology for scoring “Grounding Alignment” via LLM oracle feedback could use additional justification or possible downsides (such as bias or instability).
- **Figures A1 & A2**: These example AViRRL trajectories provide rich qualitative insight into the agent’s reasoning and its tight coupling with environment exploration decisions. However, only two example runs are shown—wider quantitative characterization of reasoning-action sequence diversity and error cases would benefit the narrative.

### Equations/Mathematical Rigor Comments

- The paper does a commendable job in formalizing the policy and hybrid action space. Equations for $\mathcal{L}_{\mathrm{SFT}}$, $\mathcal{L}_{\mathrm{GRPO}}$, and reward composition are clearly provided (Appendices A3.1, A3.2).
- However, there is some underspecification about the normalization of advantages in $\hat{A}^{(i)}=r^{(i)}-\bar{R}$, especially regarding variance reduction or stability (e.g., is there normalization across episodes/batches?). The masking of malformed/non-terminated samples is described, but the quantitative impact is not reported and should be explicitly clarified.
- Additionally, the “progress” component of the reward (Appendix A3.2) specifies only the positive direction ($\max(0, d_t-d_{t+1})$), which, while designed not to penalize exploration, might unintentionally allow for oscillatory or inefficient trajectories in the absence of a strong terminal reward. An empirical or theoretical discussion around this would strengthen the theoretical articulation.

### Potentially Missing Related Work

1. Directly tackles chain-of-thought reasoning in the parameter/latent space, relevant for the trajectory-level RL reasoning approach of AViRRL. Should be cited in Section 2 and compared with respect to reasoning granularity and performance/efficiency.
2. *AVRT: Audio-Visual Reasoning Transfer through Single-Modality Teachers*
   - Presents a supervised-to-RL transfer approach featuring efficient reasoning acquisition, which aligns with the use of warm-start via MCTS and transfer for fine-tuning seen here. This should be referenced and discussed in the Related Work and perhaps included as an additional baseline, if possible.
3. *Reasoning-Aligned Perception Decoupling for Scalable Multi-modal Reasoning*
   - Proposes perception/reasoning decoupling for multi-modal models, relevant for AViRRL’s reasoning-action separation. Please discuss in Section 2 and, if possible, compare approaches to decoupling and recombination.
4. *VFScale: Intrinsic Reasoning through Verifier-Free Test-time Scalable Diffusion Model*
   - Focused on scalable reasoning approaches that could provide alternative/orthogonal solutions to active visual reasoning. Section 2 should acknowledge and clarify distinctions.
5. *Can I Trust Your Visual Thinking? Measuring and Improving Visual Thinking Faithfulness*
   - Focused on measuring/improving faithfulness, which relates to your Grounding Alignment metric (Table 3). Should be cited in Section 5.4 and Section 2, and possibly leveraged for evaluation (or as a benchmark).
6. *RLP: Reinforcement as a Pretraining Objective*
   - Proposes RL as a general pretraining paradigm, relevant for your multi-turn RL on VLMs. Compare approaches for pretraining and discuss in both Related Work and Section 4.
7. *iGRPO: Self-Feedback–Driven LLM Reasoning*
   - Features self-feedback mechanisms in multi-turn RL, which could be an alternative to trajectory-level optimization used here. Should be cited in Section 4 and compared.
8. *SimpleTIR: End-to-End Reinforcement Learning for Multi-Turn Tool-Integrated Reasoning*
   - Presents multi-turn RL for tool-based reasoning, directly connected to multi-turn action-reasoning in AViRRL. Section 2 and Section 4 should cover this.
9. *NavA^3: Understanding Any Instruction, Navigating Anywhere, Finding Anything*
   - Proposes a hierarchical navigation framework, highly relevant to long-horizon, multi-goal embodied navigation; please situate AViRRL explicitly relative to this in Section 2.

**Questions:**

1. **Generalization Beyond TinyNav**: How does AViRRL generalize to real-world or dynamic environments, or to queries beyond category-based search (e.g., object relations, temporal sequences)?
2. **Ablation on Reward Components**: Can you provide explicit ablation studies on the weighting and inclusion of different reward components (success/format/progress)? How sensitive is performance to these values?
3. **Intervention on Reasoning**: Are there empirical experiments where reasoning is systematically ablated or perturbed to analyze the direct impact on performance? For example, what happens if the model is forced to use only shallow or template-based thoughts?
4. **Failure Modes**: Can you supply concrete examples and error analysis for common failure cases (e.g., repeated cycles/ping-ponging, confusion with distractor objects, or break-downs in grounding alignment)?
5. **Warm Start Cost**: What are the precise computational costs and time for the MCTS warm start at scale? Have you considered alternative warm-start sources, such as synthetic teachers or smaller models?

---

### Official Review · Reviewer_VxN6 · 2025-11-02

**Soundness:** 3
**Presentation:** 4
**Contribution:** 3
**Rating:** 8
**Confidence:** 3

**Summary:**

The authors present Active Visual Reasoning with Reinforcement Learning (AViRRL), a method for active visual reasoning in embodied AI environments. The key insight is to leverage pre-training to get a reasonable prior of agent behavior, followed by reinforcement learning (RL) to improve the system's reasoning capabilities in online settings. Pre-training takes the form of Monte Carlo Tree Search. The reinforcement learning phase uses a combination of sparse, formatting-based, and distance-based exploration rewards. While the overall ideas used are not new at a high level, the authors present a well-designed method and present thorough experiments. Experiments compare to zero-shot prompting methods, single-turn RL methods, and self-ablations that either discard the pre-training or RL fine-tuning, showing significant improvements over these methods.

**Strengths:**

- Overall the paper is well written, with a clear motivation, a clearly laid out description of the method, and comprehensive experiments. The analysis in the experiments is adequate.
- Strong results when combining (1) MCTS for pre-training and (2) RL fine-tuning. Ablations provide evidence that both items are critical. While the finding is not surprising per se, the method is well executed and experiments are comprehensive

**Weaknesses:**

- It would be useful to shed more details on the MCTS phase. The paper mentions that this process is expensive, but more concretely, what resources are needed for this phase, and how do they compare to the resources used for RL?
- The single-turn BC/RL methods are a bit unclear. What do they exactly do, and how are they different than multi-turn RL? Are they single-turn BC methods or RL methods? They are introduced as single-turn BC in the text, but tables 1 and 2 label them as "Single-turn RL"
- It would be interesting to see an ablation of the reward design. There are three components, sparse, formatting, and exploration rewards. How important is each component?

**Questions:**

Please see the questions raised above in the weaknesses section.

---

### Meta-Review · Area_Chair_Fiqo · 2026-01-07

**Summary:**

This paper presents AViRRL. It's a reinforcement learning approach that enables vision-language models to develop active visual reasoning strategies in 3D environments. By optimizing multi-turn thought–action trajectories, the method achieves improved performance on visual search tasks compared to existing baselines.

The paper received 8, 4, 4, 2. The reviewers have concerns about
- limited comparisons with relevant multi-turn RL and reasoning models
- limited score of benchmarking,
- limited contributions

 No rebuttals were provided. The AC has no ground to accept.

**Reviewer Concerns:**

No rebuttal were provided. All concerns regarding novelty and comparisons were not addressed.

**Reviewer Scores:**

Reviewer VxN6: 8: accept, good paper

Reviewer Jcv1: 4: marginally below the acceptance threshold.

Reviewer jCdQ: 2: reject, not good enough

Reviewer KnRF:  4: marginally below the acceptance threshold.

No rebuttal provided. AC assumes that the reviewers would not have raised their scores.

---

### Decision · Program_Chairs · 2026-01-26

Reject